# The Combination of Glucocorticoids and Hyaluronic Acid Enhances Efficacy in IL-1β/IL-17-Treated Bovine Osteochondral Grafts Compared with Individual Application

**DOI:** 10.3390/ijms241814338

**Published:** 2023-09-20

**Authors:** Christoph Bauer, Lukas B. Moser, Daniela Kern, Vivek Jeyakumar, Stefan Nehrer

**Affiliations:** 1Center for Regenerative Medicine, University for Continuing Education Krems, 3500 Krems, Austria; 2Department of Orthopedics, University Hospital Krems, 3500 Krems, Austria

**Keywords:** osteoarthritis, osteochondral graft, glucocorticoid, hyaluronic acid, IL-1β, IL-17, biotribology

## Abstract

Patients with knee osteoarthritis often receive glucocorticoid (GC) or hyaluronic acid (HA) injections to alleviate symptoms. This study evaluated the impact of Triamcinolone Hexacetonide (a GC), HA, and a combination of both on bovine osteochondral grafts exposed to IL-1β and IL-17 in an ex vivo culture. Metabolic activity increased with GC treatment. GCs and GCs/HA counteracted cytokine effects, with gene expressions similar to untreated controls, while HA alone did not. However, HA improved the coefficient of friction after two weeks. The highest friction values were observed in GC-containing and cytokine-treated groups. Cytokine treatment reduced tissue proteoglycan content, which HA could mitigate, especially in the GC/HA combination. This combo also effectively controlled proteoglycan release, supported by reduced sGAG release. Cytokine treatment led to surface cell death, while GCs, HA, or their combination showed protective effects against inflammation. The GC/HA combination had the best overall results, suggesting its potential as a superior treatment option for osteoarthritis.

## 1. Introduction

Osteoarthritis (OA) is a prevalent degenerative joint disease affecting hundreds of millions of people worldwide [1]. Its etiology involves multiple factors, including biochemical imbalances, surface degradation due to mechanical stress, osteophyte formation, subchondral bone remodeling, and joint inflammation [2]. Inflammatory responses in the joint lead to the release of pro-inflammatory cytokines (e.g., IL-1β, TNF-α, IL-6, or IL-17) and proteolytic mediators (e.g., MMPs). This results in disrupted metabolic homeostasis, cartilage tissue degradation, chondrocyte apoptosis, reduced synthesis of crucial extracellular matrix (ECM) components [3,4], and increased coefficient of friction (COF) [5].

Extensive research has focused on chondroprotective approaches to delay surgery and preserve joint function. Although lifestyle changes, physiotherapy, and pain medication are commonly used in early knee OA, they merely offer symptomatic relief without cartilage protection [6]. Similarly, the minimally invasive subchondroplasty technique provides pain relief and aids in early-stage osteoarthritis (OA) management by facilitating bone repair beneath the affected cartilage, yet it lacks a chondroprotective effect [7]. In advanced stages of OA, where cartilage wear is severe and bone exposure occurs, knee arthroplasty remains the preferred treatment method. However, exploring chondroprotective approaches aims to delay the need for surgery. Intra-articular injections of various medications promise to achieve localized effects on chondrocytes [8]. Glucocorticoids (GCs), for instance, are potent in alleviating inflammation and pain, but their prolonged use might induce pro-apoptotic effects on chondrocytes [9].

In contrast, hyaluronic acid (HA) has gained popularity due to its ability to reduce inflammation, enhance lubrication, and improve viscoelastic properties by stimulating endogenous HA synthesis. In osteoarthritic chondrocytes, HA promotes cellular metabolic activity and biosynthesis, reducing pain and improving patient function. Recent efforts have explored the combination of GCs and HA to achieve synergistic chondroprotective effects [10,11].

Furthermore, various studies have investigated changes in cartilage biomechanical and biotribological properties, exploring the impact of degradative enzymes and depletion of glycosaminoglycans from the cartilage matrix [12,13]. Understanding the role of inflammation in altering the frictional properties of cartilage may unveil novel strategies to suppress joint degradation.

While clinical studies have reported decreased pain levels in knee OA patients with the combined administration of GCs and HA compared to separate ones [14,15], experimental studies are lacking to confirm these beneficial effects at the cellular level. Therefore, our study aims to address this crucial knowledge gap by examining the effects of these drugs on osteochondral grafts.

This study investigates the influence of GCs, HA, and their combination on osteochondral grafts treated with pro-inflammatory cytokines, assessing their effects on biological and biotribological outcome measures. We hypothesize that the combined treatment product offers superior chondroprotective properties compared to the singular administration of GCs or HA.

## 2. Results

### 2.1. Harvesting of Osteochondral Grafts

Osteochondral grafts were carefully harvested from the medial condyle of bovine knee joints, ensuring a planar, parallel cartilage surface, which was critical for the precision of tribological tests. Although some asymmetric grafts were inevitable, their use was restricted to control and treatment groups where surface flatness was not crucial. Figure 1 visually represents the osteochondral grafts after the extraction and cutting process.

### 2.2. Metabolic Activity of Chondrocytes

As represented in Figure 2, the metabolic activity of chondrocytes within the control group of osteochondral grafts displayed a close resemblance to the metabolic activity observed in cytokine-treated, +HA, and +GC/HA groups. Notably, the chondrocytes in the +GC group manifested a statistically significant increase in metabolic activity compared to those in the control group.

### 2.3. Expression of Anabolic and Catabolic Genes

We designed specific bovine primers to assess gene expression and successfully refined the annealing temperature values. After a 14-day incubation period, chondrocytes from osteochondral grafts treated with either GCs or GCs/HA exhibited expression patterns akin to those of the untreated control group. Additionally, the expression patterns of the cytokine and HA-treated groups were also similar. Crucially, the GC or GC/HA treatment notably enhanced the expression of the anabolic genes COL2A1 and ACAN, as depicted in Figure 3A,B. In comparison, HA treatment did not significantly influence the expression of these genes.

Conversely, the GC and GC/HA treated groups revealed a decline in gene expression for the non-cartilage-specific gene COL1A1 (Figure 3C) and for the genes encoding the degradative enzymes MMP1 and MMP13 (Figure 3D,E). Gene expression levels in the control group were in line with those of the GC and GC/HA groups for these latter genes. Significantly, the GC/HA group exhibited a notable decrease in the expression of MMPs and COL1A1 compared to the cytokine and HA groups, a change not observed in the GC group alone.

### 2.4. Live/Dead Staining

Following the 14-day incubation period, osteochondral grafts were stained with Calcein AM and Ethidium Homodimer-1 to visually represent the distribution of live and dead cells within the cartilage tissue. As per Figure 4, the cytokine-only treated group displayed a layer of dead cells on the surface, a feature not observed in the other groups.

### 2.5. Histology

We performed Safranin O/Light Green staining, as depicted in Figure 5, to evaluate potential variances in proteoglycan content and surface roughness between untreated (control) and cytokine-treated osteochondral grafts (with or without GC, HA, or GC/HA supplementation) before biotribological tests. Figure 5 demonstrates that the proteoglycan content reached its zenith in both the control and cytokine-treated groups supplemented with GCs/HA. In contrast, groups supplemented with either GCs or HA exhibited a decline in proteoglycan content, primarily in the superficial layer (visible as bright areas). A notably significant decrease was witnessed exclusively in the cytokine-treated group without any supplementation, underlining the profound influence of inflammatory mediators. Alongside the fluctuations in proteoglycan content, changes in the cartilage surface were discernible in the cytokine-treated group without supplementation and those supplemented with GCs, as evidenced by a slight surface roughening. In contrast, the control group and groups supplemented with HA (both HA and GCs/HA) maintained a smooth cartilage surface without significant changes.

### 2.6. Coefficient of Friction (COF)

Throughout the biotribological testing phase, the COF was continuously recorded across four cycles, as shown in Figure 6. Across all conditions, the COF value did not display significant fluctuations between the cycles. Interestingly, introducing HA to cytokine-treated osteochondral (OC) grafts decreased the friction coefficient by approximately 50%, matching the level observed in the untreated control group. Contrarily, the GC/HA group demonstrated an elevated COF level, 1.5 times higher than that in the cytokine and GC groups. Nevertheless, it is noteworthy that none of the experimental groups reached the COF level typically seen in the knee joint.

### 2.7. sGAG in the Supernatant

Following a 14-day incubation period, the supernatant from each sample was collected and assessed for the presence of sulfated glycosaminoglycans (sGAGs), as visualized in Figure 7. The detected quantities of sGAGs showcased a high correlation with the Safranin O staining results. The untreated group and the group supplemented with GCs/HA demonstrated the smallest numbers of sGAGs in the supernatant but exhibited the most intense Safranin O staining in the histological samples. The remaining conditions aligned with these histological findings, whereby the cytokine-treated osteochondral (OC) grafts displayed increased release of sGAGs into the supernatant, contrasted with an evident decrease of sGAG content within the tissue. Analogous results were seen in the groups treated with GCs and HA.

## 3. Discussion

This study demonstrates that the combination of hyaluronic acid (HA) and the glucocorticoid (GC) Triamcinolone Hexacetonide effectively counteracts osteoarthritic conditions. Specifically, the combination product showed an overall positive effect on cell viability and expression of anabolic and catabolic genes, and it mitigated the release of sulfated glycosaminoglycans (sGAGs). The investigation utilized an ex vivo culture model, where osteochondral grafts were pre-treated with cytokines to induce a pro-inflammatory state before administering the combination product, GCs, and HA into the culture medium. The grafts were sourced from the medial femoral condyle of bovine cartilage to ensure consistency and accessibility.

The utilization of HA in clinical practice for osteoarthritis treatment has previously evidenced its capacity to bolster the lubricating and shock-absorbing features of synovial fluid [16]. Additionally, HA can enact disease-modifying effects such as inflammation reduction [17] or pain relief, with concomitant safe use, even with multiple administrations [18]. GCs, despite their propensity to trigger chondrocyte apoptosis [19,20], are also applied in clinical practice for osteoarthritis treatment due to their anti-inflammatory effects. When combined with HA, the goal is to diminish the adverse effects of GCs while retaining their anti-inflammatory properties, alongside HA’s lubricating and shock-absorbing attributes. The anti-inflammatory potentials of this combination product have been validated in a study involving 2D-cultured osteoarthritic chondrocytes [21]. Utilizing this ex vivo model with bovine osteochondral grafts is intended to emulate a more native model. Initially, the harvested grafts were exposed to cytokines for three days to initiate an inflammatory process akin to that seen in osteoarthritis [22]. Subsequently, to simulate the intra-articular injection treatment of osteoarthritis as closely as possible, the test substances were directly introduced into the cytokine-laden culture medium.

Bovine osteochondral grafts were harvested in two different diameters to negate potential edge effects that may occur, such as sliding along the edges during tribological tests. The top 8 mm graft was consistently positioned on the surface of the lower 10 mm graft. This cartilage-on-cartilage testing system efficiently simulates physiological conditions, resulting in lower contact stress and frictional forces than a metal-on-cartilage system [23]. This setup is beneficial for emulating inflammatory reactions, as the outcomes depend not solely on the test fluid but also on the cartilage’s response to treatment. Additionally, the applied pressure of 3.57 MPa is within the range observed during pressure measurements of the tibial–femoral compartment in a human knee joint under typical physical load [24].

To establish osteoarthritic (OA) conditions, we selected the cytokines IL-1β and IL-17 for use in this study due to their critical role in the onset and progression of osteoarthritis. IL-1β activates other pro-inflammatory molecules and enzymes, such as IL-6 or TNF-α, which play crucial roles in OA pathogenesis. These factors’ activation is further supported by IL-17, which encourages cartilage degradation and joint damage within the broader system. In osteoarthritis, these cytokines exist in picogram ranges, driving a continuous inflammatory process over several months and years, eventually contributing to cartilage degradation during OA’s pathogenesis [22,25]. The elevated cytokine concentration used in this study was designed to simulate the state of osteoarthritis as rapidly as possible, as it has been demonstrated to enhance the expression of genes associated with degradative processes [26,27].

Exposing the osteochondral grafts to the cytokine mixture led to a slight uptrend in metabolic activity, although the increase was statistically insignificant. This trend was also seen under HA or GC/HA conditions. However, introducing GCs resulted in a significant increase in metabolic activity. This result likely arises from GCs’ stimulatory effects on chondrocyte proliferation and glycolysis and reduced chondrocyte metabolic inhibition due to GCs’ anti-inflammatory action [28]. A study by Sherman et al. demonstrated that GCs (specifically Triamcinolone acetonide) positively affect the metabolic activity of chondrocytes in cartilage tissue [29]. Even though this impact was observed without cytokine treatment, it supports our findings. However, a separate study using a 2D culture did not report increased metabolic activity with the GC Triamcinolone Hexacetonide [21]. This discrepancy might stem from the earlier study’s use of human osteoarthritic chondrocytes, whereas the present study utilized bovine chondrocytes. The contrast between 2D and tissue cultures likely contributes significantly to the difference.

The study mentioned above by Sherman et al. also demonstrates that GCs do not negatively impact cell viability [29]. In our study, the control and supplement groups showed no evidence of cell death on the cartilage surface compared to the cytokine-treated osteochondral grafts. Interestingly, while the application of HA did not significantly affect cell viability, both GCs and the combination product showed a positive effect on cell viability. This observation could be related to the anti-inflammatory properties of GCs [28], which help mitigate inflammation-induced cell death [30]. In contrast to our results, a study by Yan et al. reported an increase in chondrocyte apoptosis following GC treatment [20]. This discrepancy may be due to differences in the GC used (Dexamethasone was used in their study, while Triamcinolone Hexacetonide was used in ours) and the use of 2D-cultured cells in their study compared to our tissue culture approach.

Furthermore, the GC concentration used in this study at 5 mg/mL is similar to the concentration we used, 4.5 mg/mL. HA reduced the loss of proteoglycans noted in both the standalone treatment and the combined product. Furthermore, an in vivo study by Tschon et al. verified that proteoglycan levels remain elevated when using a combination of HA and GCs [31]. As described earlier, surface changes instigated by pro-inflammatory cytokines have been substantiated by various studies [32,33]. This led to doubling the coefficient of friction (COF) relative to the control group. Only HA succeeded in preserving the COF level of the control group, whereas GCs (COF ~ 0.08) and GCs/HA (COF ~ 0.12) resulted in significantly higher values. The COF values for all groups exceeded the physiological range [34] and the range observed in a study conducted by Farnham and Price (2020) using synovial fluid [35]. However, the latter study did not employ a cartilage-on-cartilage model. Higher values are likely due to our use of Phosphate Buffered Saline (PBS), which is not a lubricant and had a COF of 0.16 in a study by Rebenda et al. using porcine osteochondral grafts [36]. Nevertheless, we used PBS to discern the effects of different treatments on COF, not to measure the direct influence of the supplement on COF. Therefore, the supplement’s effect on COF cannot be accurately gauged at this juncture, even though the osteochondral grafts were flushed with PBS before tribological testing.

Introducing pro-inflammatory cytokines into the culture medium of osteochondral grafts resulted in a decline in the expression of the anabolic genes collagen type 2 and aggrecan compared to the control group. Additionally, genes expressing degradative enzymes were significantly elevated. These observations are consistent with the literature [25,37] and suggest that the treatment induces conditions typically present in osteoarthritis. HA supplementation did not elicit any changes, even though high-molecular-weight HA typically exhibits anti-inflammatory effects [38,39], as previously demonstrated in one of our studies [17]. Adding GCs to cytokine treatment (GCs and GCs/HA) attenuated the negative impacts of inflammation, leading to increased expression of anabolic genes and reduced expression of catabolic genes. Pemmari et al. reported a similar effect by incubating osteoarthritic cartilage tissue with GCs for 24 h [40]. Including GCs led to increased aggrecan gene expression and downregulated the expression of metalloproteinases 1 and 13. Changes induced by adding GCs to the cytokine-infused culture medium yielded expression levels of anabolic and catabolic genes comparable to those in the control group.

During the two-week incubation of osteochondral grafts with pro-inflammatory cytokines and supplements, sulfated glycosaminoglycans (sGAGs) were released into the culture medium. These were analyzed following a lyophilization step, which was undertaken to allow for the resuspension of the content in a smaller volume. IL-1β and IL-17 in the culture medium significantly increased the release of sGAGs compared to the control group, indicating cartilage degradation. Similar sGAG release has been reported in other studies using IL-1β or TNF-α [41,42]. Consistent with our study, it has also been noted that adding a GC mitigated the release of sGAGs from the tissue [41,43]. Incorporating HA or the combination product further diminished this release, and all values correlated with the histology data.

However, our study has several limitations that need to be considered. Firstly, it is important to acknowledge the limited sample size in this study. As a result, the findings can only suggest a preliminary trend, and further research is needed to validate these results. Secondly, the limited number of cytokines used can be perceived as a restriction. During the development of osteoarthritis, many inflammatory factors are discharged into the synovial fluid due to an imbalance between pro-inflammatory and anti-inflammatory agents [44]. Moreover, different combinations of cytokines have been employed in other studies, which could lead to a varied inflammatory response [26,45]. Further, cartilage damage would typically occur over a much longer timeframe. Another limiting factor is using Phosphate Buffered Saline (PBS) as the test fluid, which is not a lubricant, although it is frequently used in tribological tests [23]. Despite PBS being a lubricant with a friction coefficient ten times higher than synovial fluid (SF) or HA in a steady state [35], we chose PBS for our tribological tests to minimize variables (e.g., differences in SF composition). In addition to the mentioned intra-articular injections, others such as platelet-rich plasma (PRP), autologous conditioned serum (ACS), or mesenchymal stem cell injections (MSCs) could have been examined. But this is already planned for a subsequent study. Lastly, the model itself can be considered a limitation, as it is an ex vivo model, and its comparison with physiological conditions is challenging, given that the absence of surrounding tissue can lead to abnormal loading conditions.

## 4. Materials and Methods

### 4.1. Specimen Preparation and Study Design

Five bovine knees were obtained from cows aged between 18 and 20 months from a local slaughterhouse. Osteochondral grafts were harvested from the medial femoral condyle under aseptic conditions using a Single-Use OATS punch (Arthrex Inc., Naples, FL, USA). Each knee yielded between 24 and 28 osteochondral grafts with a diameter of 8 mm. Additionally, a separate bovine knee from the same cow was used to obtain osteochondral grafts with 10 mm diameters, specifically for biotribological tests. After harvesting, all grafts were standardized to a length of 8 mm.

To ensure cleanliness and removal of loose bone particles and fatty tissue, the osteochondral grafts were thoroughly washed in Phosphate Buffered Saline (PBS, Sigma-Aldrich Chemie GmbH, Steinheim, Germany). Subsequently, they were transferred to a 12-well plate containing growth medium (GIBCO^®^ DMEM/F12 GlutaMAXTM-I, Life Technologies, Carlsbad, CA, USA) supplemented with 10% Fetal Calf Serum (FCS; GIBCO^®^, Life Technologies, Carlsbad, CA, USA), antibiotics (penicillin 200 U/mL; streptomycin 0.2 mg/mL), Amphotericin B 2.5 µg/mL (Sigma-Aldrich Chemie GmbH, Steinheim, Germany), and 0.05 mg/mL ascorbic acid (Sigma-Aldrich Chemie GmbH, Steinheim, Germany).

As shown in Figure 8, the osteochondral grafts were incubated under controlled conditions in the growth medium for three days. After this initial incubation period, the culture conditions were modified using 5% FCS in the growth medium, and further incubation for an additional three days was conducted.

Following the six-day preincubation, the osteochondral grafts were transferred to 25 mL cell culture flasks containing growth medium supplemented with 5% FCS and pro-inflammatory cytokines IL-1β and IL-17 (both at a concentration of 10 ng/mL, Sigma-Aldrich Chemie GmbH, Steinheim, Germany). The grafts were then incubated for three days under these conditions.

In the next step, to the initial treatment, 10% of the test substances (glucocorticoids (GCs), hyaluronic acid (HA), and glucocorticoids/hyaluronic acid combination (GCs/HA)) were added to the respective flasks, and further incubation was carried out for 11 days. Alongside these three treatment conditions, osteochondral grafts were also cultured with and without the supplementation of cytokines alone (Table 1).

After two weeks of incubation, the cartilage and chondrocytes of the osteochondral (OC) grafts were analyzed under various culture conditions.

### 4.2. Metabolic Activity

An XTT-based ex vivo toxicology assay kit (Cell Proliferation Kit II, Roche Diagnostics, Basel, Switzerland) was utilized following the manufacturer’s instructions to assess the metabolic activity of chondrocytes within the tissue.

The cartilage was carefully excised from the osteochondral grafts using a scalpel and then longitudinally divided into two parts for subsequent analysis. One part was designated for the XTT assay, while the other was intended for RNA isolation. The cartilage was minced into smaller fragments on a 24-well plate. After determining the tissue weight for each sample, the tissue was incubated in an XTT solution containing 1 mL medium, 490 µL XTT reagent, and 10 µL activation reagent. The incubation was carried out for four hours at 37 °C in an atmosphere containing 5% (*v*/*v*) CO_2_. After the incubation period, the XTT solution was carefully removed and retained. The remaining tetrazolium product within the tissue was extracted by incubating it with 0.5 mL dimethyl sulfoxide (DMSO) for one hour at room temperature with continuous agitation. The XTT and DMSO solutions were then combined, and the absorbance was measured at 492 nm, with a background wavelength of 690 nm, in triplicate using a multi-mode microplate reader (SynergyTM 2, Winooski, VT, USA) equipped with Gen 5 software. The absorbance values were normalized to the wet weight of the tissue.

### 4.3. RNA Isolation

The second half of the cartilage tissue obtained from the OC grafts was preserved in RNAlater™ (Qiagen, Hilden, Germany) at 4 °C for up to one week. After the storage period, the cartilage was minced into smaller fragments and transferred to tubes containing ceramic beads (MagNA Lyser Green Beads, Roche Diagnostics, Basel, Switzerland) along with 300 µL lysis buffer (10 µL ß-mercaptoethanol + 290 µL RLT from the Fibrous Tissue Kit, Qiagen, Hilden, Germany). The tubes were stored in liquid nitrogen until RNA isolation. For the isolation process, the tubes were thawed and placed in the MagNA Lyser (Roche Diagnostics, Basel, Switzerland) for homogenization of the cartilage tissue. Homogenization was performed at 6500 rpm for 20 s, with four repetitions and a 2 min cooling phase after each step. Following the manufacturer’s instructions, each sample was incubated with 20 µL Proteinase K (from the Fibrous Tissue Kit) for 30 min to enhance RNA yield. Finally, RNA was eluated in 30 µL and stored at −80 °C until cDNA synthesis.

### 4.4. Gene Expression Analysis

Gene expression analysis was conducted as previously described [46]. Briefly, cDNA synthesis was performed using the Transcriptor First Strand cDNA Synthesis Kit (Roche, Basel, Switzerland). Additionally, RNA from bacteriophage MS2 was added during cDNA synthesis to stabilize the isolated RNA. Subsequently, a real-time quantitative polymerase chain reaction (RT-qPCR) was conducted in triplicate using the LightCycler^®^ 96 instrument from Roche (Basel, Switzerland). Four genes, including Collagen type 2 (COL2A1), Aggrecan (ACAN), Matrix Metalloproteinase-1 (MMP1), and Matrix Metalloproteinase-13 (MMP13), were analyzed, with Glyceraldehyde-3-phosphate dehydrogenase (GAPDH) serving as a housekeeping gene (Table 2).

### 4.5. Histology

For histological analysis, the osteochondral (OC) grafts were fixed in a 4% buffered formaldehyde solution (VWR, Radnor, PA, USA) for up to 1 week. Decalcification was then performed under constant agitation using the Osteosoft solution (Merck, Burlington, MA, USA) for 4 to 6 weeks. Following decalcification, the OC grafts were embedded in Tissue-Tek^®^ OCT (Optimal Cutting Temperature, VWR, Radnor, PA, USA) and stored at −80 °C. Sectioning was performed using the CryoStarTM NX70 Cryostat (Thermo Fischer Scientific, Waltham, MA, USA) with a knife temperature of −25 °C and a chamber temperature of −20 °C. Then, 6 µm sections were obtained and subjected to Safranin O/Light Green staining. Images were captured using a Leica DM-1000 microscope and processed using Leica IM500 Image Manager software Version 5 (Leica, Wetzlar, Germany).

### 4.6. Live/Dead Staining

A thin section of the divided cartilage, previously used for metabolic activity and gene expression analysis, was obtained using a microtome blade and utilized for live/dead staining. The sections were placed in a 12-well plate and washed once with Dulbecco’s Phosphate Buffered Saline (DPBS). The staining solution was prepared by combining 1 µM Ethidium Homodimer-1 and 1 µM Calcein AM in DPBS. Then, 500 µL of the staining solution was added to each well, and incubation was performed at 37 °C for 1 h. After incubation, the sections were washed three times with DPBS and transferred onto glass slides. A drop of mounting medium ProLongTM antifade reagent (Invitrogen, LifeTech Austria, Vienna, Austria) was applied to each section then covered with a cover glass. Live and dead cells within the tissue were imaged using a confocal microscope (Leica TCS SP8 MP, Leica Microsystems, Wetzlar, Germany).

### 4.7. Biotribological Test System

The biotribological tests were conducted using a tribometer from Rtec Instruments, Inc. (San Jose, CA, USA) with custom-made holders designed specifically for the OC grafts (Figure 9). Before testing, the OC grafts were rinsed with PBS to remove residual supplements. The grafts with a diameter of 8 mm were fixed on the top, while those with a diameter of 10 mm were fixed on the bottom. During the tests, the OC grafts were immersed in PBS and subjected to reciprocal motion against each other. An applied load of 180 N was used, resulting in an estimated average pressure value of 3.57 MPa considering the contact area of 50.26 mm^2^ for the ∅8 mm cartilage samples. The plane was assumed to be flat for all tested cartilage samples. To simulate the loading and unloading conditions experienced by the knee during walking, the system was loaded for ten minutes and then unloaded for another five minutes, resulting in a total test period of one hour (4 cycles). The stroke length was set at 2 mm, and the frequency was set at 1.5 Hz. The established tribo-model closely matched the contact pressure measurements of the tibiofemoral compartment of a human knee under the average body weight load with 0° flexion [47]. All tests were conducted at room temperature.

### 4.8. Sulfated Glycosaminoglycans (sGAGs)

Following the two-week osteochondral (OC) graft incubation period, the culture medium from each experimental condition was collected, frozen, and subsequently lyophilized. The resulting lyophilizate was resuspended in a culture medium (without FCS) and used to quantify sulfated glycosaminoglycans (sGAGs) following the protocol outlined by Barbosa et al., 2003 [48].

In brief, the resuspended lyophilizate was subjected to overnight treatment with 25 U/mL proteinase K (Sigma-Aldrich, St. Louis, MO, USA) at 56 °C. Following enzyme inactivation (90 °C, 10 min), the fluid was collected in ultra-free filter reaction tubes with a pore size of 0.1 µm (Millipore, Burlington, MA, USA) and centrifuged (12,000× *g*, 4 min, room temperature). Subsequently, 1 mL of a 1,9-dimethyl-methylene blue solution (DMMB) was added to 100 µL of the filtrate and vigorously mixed to facilitate the formation of complexes between DMMB and sGAGs in the sample. The complexes were pelleted via centrifugation (12,000× *g*, 10 min, room temperature) and dissolved in a decomplexation solution. After 30 min of shaking, the absorbance was measured at 656 nm using a photometric Ultrospec 3300 pro spectrophotometer (Amersham Bioscience plc, Amersham, UK). The sGAG content was determined based on a standard curve generated using shark chondroitin sulfate (Sigma-Aldrich, St. Louis, MO, USA) as a reference. The sGAG measurements for all experimental conditions were performed in duplicate.

### 4.9. Statistical Analysis

All statistical analyses were conducted using GraphPad Prism Software Version 9 (GraphPad Prism Software Inc., San Diego, CA, USA). A one-way ANOVA was employed for the statistical analysis, followed by multiple comparisons using a non-parametric Kruskal–Wallis test and Dunn’s post hoc test. Data from metabolic activity and gene expression are presented using box plots, illustrating the median, first quartile, and third quartile, with error bars representing the maximum and minimum values. The coefficient of friction values are reported as means. Statistical significance was set at *p* < 0.05.

## 5. Conclusions

The study revealed that the supplementation of GCs, HA, or a combination of both had beneficial effects on inflammatory conditions in this ex vivo model. Notably, the combined GC/HA product demonstrated superior outcomes in most tests, as evidenced by the gene expression data for anabolic and catabolic markers, tissue cell viability, and proteoglycan presence in histological sections. Furthermore, the release of sGAGs was notably reduced with the GC/HA combination. However, the only drawback observed was a higher coefficient of friction compared to other groups. But the additionally performed tribological analysis, along with other analytical methods, offers new insights into the different supplements. These results can offer crucial recommendations to orthopedic surgeons when choosing intra-articular substances to treat osteoarthritis.

## Figures and Tables

**Figure 1 ijms-24-14338-f001:**
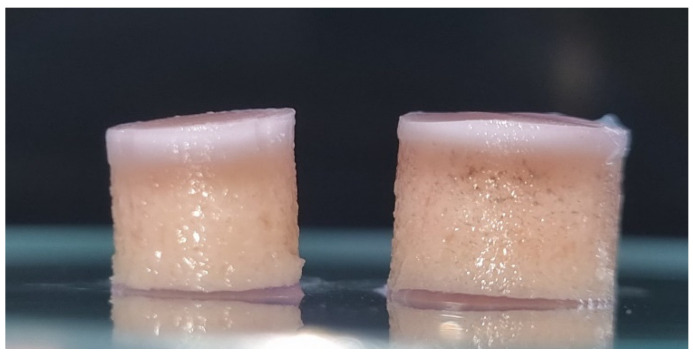
Visual depiction of osteochondral grafts post-extraction and trimmed to the required length, with samples measuring 8 mm (**left**) and 10 mm (**right**), respectively.

**Figure 2 ijms-24-14338-f002:**
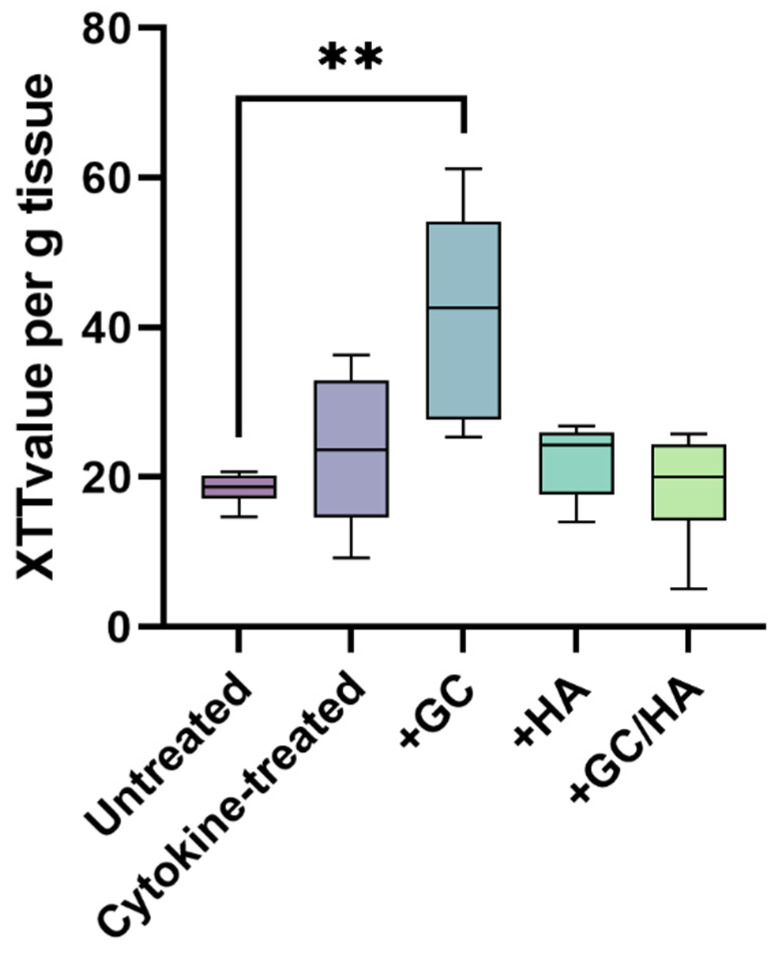
A comparative illustration of the metabolic activity (assessed using the XTT assay) of chondrocytes in osteochondral (OC) grafts, representing both untreated and cytokine-treated grafts (supplemented or not). The double asterisk denotes a statistically significant difference, with *p* < 0.01.

**Figure 3 ijms-24-14338-f003:**
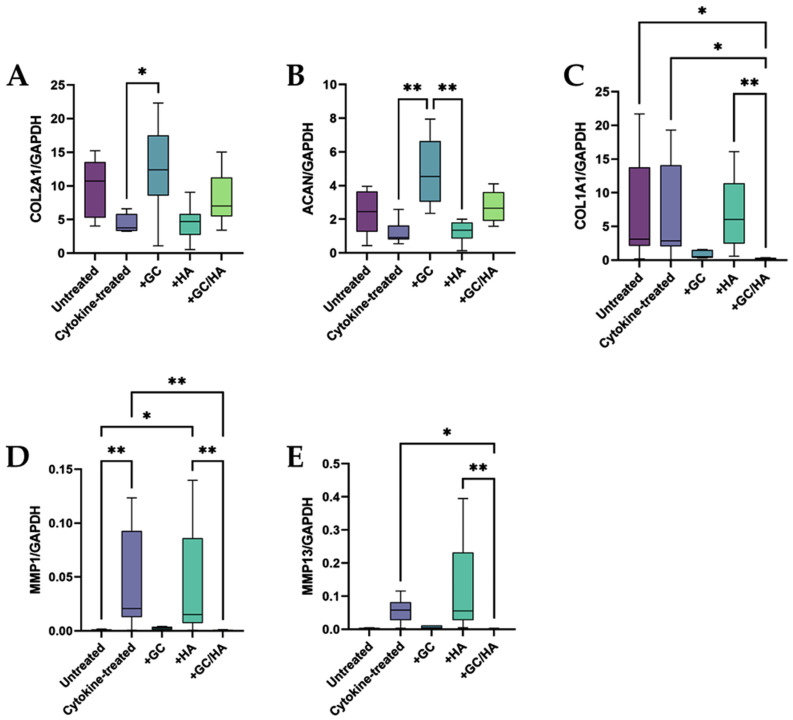
This figure represents the gene expression analysis of anabolic (**A**,**B**), catabolic (**D**,**E**), and non-cartilage-specific (**C**) genes. A single or double asterisk signifies a statistically significant difference, with *p*-values less than 0.05 or 0.01.

**Figure 4 ijms-24-14338-f004:**
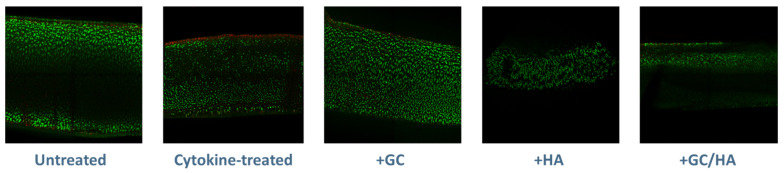
Illustration of live/dead staining results for cartilage tissue samples (approximately 200 µm in thickness), conducted with Calcein AM and Ethidium Homodimer-1.

**Figure 5 ijms-24-14338-f005:**
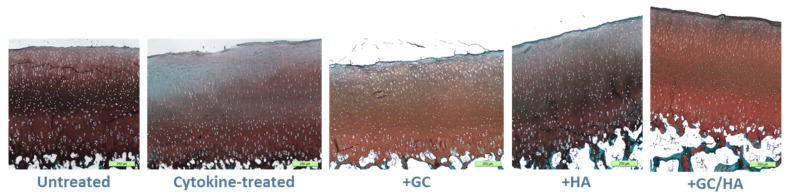
This illustration depicts representative histological cross-sections of osteochondral (OC) grafts. Both untreated and cytokine-treated samples (with and without supplements) are included, each stained with Safranin O/Light Green to highlight differences. The scale bar corresponds to 250 µm.

**Figure 6 ijms-24-14338-f006:**
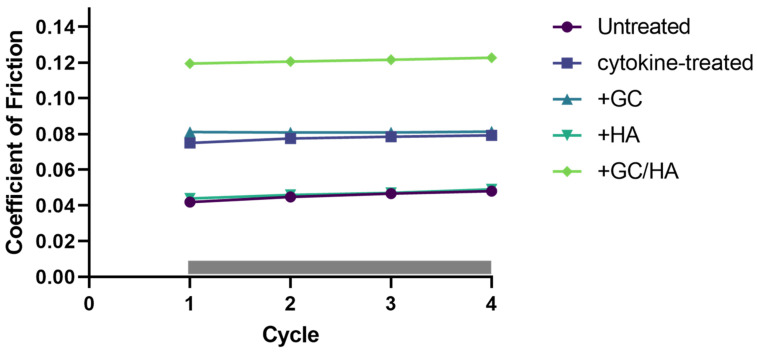
This figure captures the continuous recording of the coefficient of friction (COF) throughout the biotribological testing phase. Mean COF values were calculated for each of the four cycles, conducted over a testing period of one hour. The study involved a sample size of N = 5.

**Figure 7 ijms-24-14338-f007:**
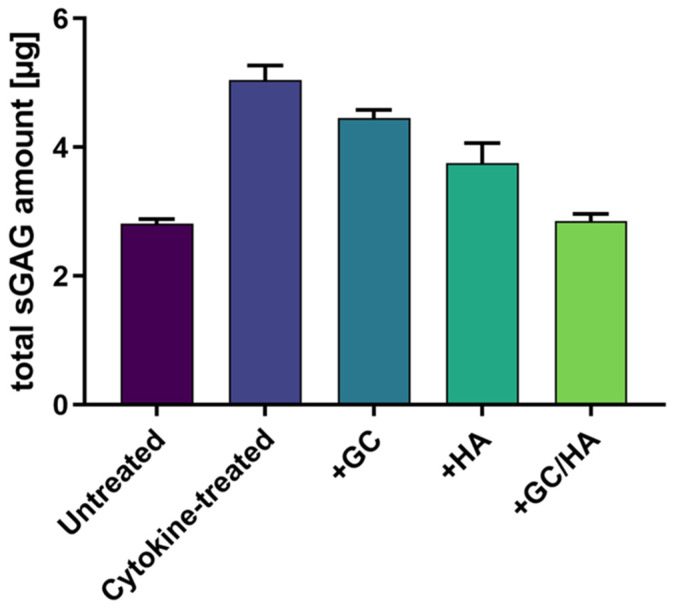
The quantification of sulfated glycosaminoglycans (sGAGs) determined in the gathered supernatants. The sample size for this analysis was N = 5.

**Figure 8 ijms-24-14338-f008:**
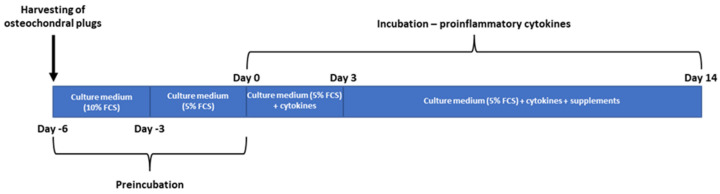
Timeline of experimental setup.

**Figure 9 ijms-24-14338-f009:**
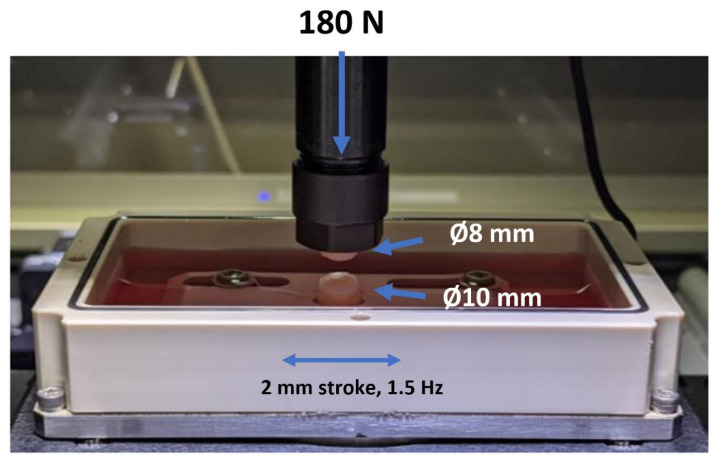
The biotribological test system’s pin-on-pin setup shows how OC grafts are fixed in the upper and lower graft holder. The upper graft holder applied the force of 180 N, while the lower holder moved with a stroke of 2 mm and 1.5 Hz.

**Table 1 ijms-24-14338-t001:** Culture conditions for osteochondral grafts.

No. of Condition	Medium Composition
1	Growth Medium
2	Growth Medium + Cytokines
3	Growth Medium + Cytokines + 10% GC [4.5 mg/mL]
4	Growth Medium + Cytokines + 10% HA [22 mg/mL]
5	Growth Medium + Cytokines + 10% GC/HA [4.5 mg/mL and 22 mg/mL]

**Table 2 ijms-24-14338-t002:** Sequences of primers and conditions used in RT-qPCR.

Primer	Abreviation	Sequence (3′-5′)
**Glyceraldehyde-3-phophate dehydrogenase**	GAPDH	
Forward		ATGTTCCAGTATGATTCCACCC
Probe		AGCTTCCCGTTCTCTGCCTTGAC
Reverse		ATACTCAGCACCAGCATCAC
**Aggrecan core protein I**	ACAN	
Forward		ACCTACGATGTCTACTGCTACG
Probe		AGAAGGTGAACTGCTCCAGGCG
Reverse		AGAGTGGCGTTTTGGGATTC
**Collagen type II, alpha I**	COL2A1	
Forward		GTGCAACTGGTCCTCTGG
Probe		CCTTGTTCGCCTTTGAAGCCAGC
Reverse		ACCTCTTTTCCCTTCTTCACC
**Matrix metalloproteinase 1**	MMP1	
Forward		TTCAACCAGGTGCAGGTATC
Probe		AAATTCATGCGCTGCCACCCG
Reverse		AGCCCCAATGTCA
**Matrix metalloproteinase 13**	MMP13	
Forward		CTAAACATCCCAAAACGCCAG
Probe		CCCTTGATGCCATAACCAGTCTCCG
Reverse		ACAGCTCTGCTTCAACCT

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
