# Peer review of "The Combination of Glucocorticoids and Hyaluronic Acid Enhances Efficacy in IL-1β/IL-17-Treated Bovine Osteochondral Grafts Compared with Individual Application"

_ijms, 2023, doi:10.3390/ijms241814338_

Round 1

Reviewer 1 Report

1. How long do Hyaluronic Acid injections in the knee last?

2. What is the efficacy and safety of Hylan versus Hyaluronic acid in the treatment of knee osteoarthritis?

3. How often can you get Hyaluronic acid injections in your knees?

4. What happens when you get Hyaluronic acid injections in your knees?

It can be improved.

Reviewer 2 Report

Dear Authors,

your paper seems well structured but some concerns have to be better addressed.

Your hypothesis is interesting and could offer new perspectives in OA treatments.

Nevertheless, you should revise references and sources along all the text. 

I suggest also to use further keywords different from those used in the title in order to improve your paper's visibility.

 What is most important, ethical concerns are not discussed. Did you receive an ethical approval? If not, please specify why. Did you register your protocol? Did you follow Helnsinki Declaration? Please clarify all these aspects.

Then, the sample size is too limited to obtain general conclusions, so I suggest to reconsider your final statements.

Similarly, the discussion should be briefly expanded. You should compare your findings with well structure in vivo studies both on animals and on humans. To do that I suggest the following recent references:

-Farì, G., Megna, M., Scacco, S., Ranieri, M., Raele, M. V., Chiaia Noya, E., Macchiarola, D., Bianchi, F. P., Carati, D., Panico, S., Di Campi, E., Gnoni, A., Scacco, V., Inchingolo, A. D., Qorri, E., Scarano, A., & Rapone, B. (2023). Hemp Seed Oil in Association with β-Caryophyllene, Myrcene and Ginger Extract as a Nutraceutical Integration in Knee Osteoarthritis: A Double-Blind Prospective Case-Control Study. Medicina (Kaunas, Lithuania)59(2), 191. https://doi.org/10.3390/medicina59020191

-Della Tommasa S, Winter K, Seeger J, Spitzbarth I, Brehm W, Troillet A. Evaluation of Villus Synovium From Unaffected Metacarpophalangeal Joints of Adult and Juvenile Horses. J Equine Vet Sci. 2021 Jul;102:103637. doi: 10.1016/j.jevs.2021.103637. Epub 2021 Apr 29. PMID: 34119205.

Farì G, Megna M, Scacco S, Ranieri M, Raele MV, Noya EC, Macchiarola D, Bianchi FP, Carati D, Gnoni A, et al. Effects of Terpenes on the Osteoarthritis Cytokine Profile by Modulation of IL-6: Double Face versus Dark Knight? Biology. 2023; 12(8):1061. https://doi.org/10.3390/biology12081061

Strootmann, T., Spitzbarth, I., Della Tommasa, S., Brehm, W., Köller, G., & Troillet, A. (2022). Synovial Fluid Analysis and Microscopic Assessment of Macrophage Quantities and Morphology in Equine Septic Arthritis. Analyse von Synovialpunktaten und mikroskopische Beurteilung der phänotypischen Ausprägung von Makrophagen bei Pferden mit septischer Arthritis. Tierarztliche Praxis. Ausgabe G, Grosstiere/Nutztiere50(6), 377–385. https://doi.org/10.1055/a-1956-5245

Best regards and good luck

Reviewer 3 Report

I commend the authors for their research entitled "Impact of hyaluronic acid and glucocorticoids on cartilage and friction in IL-1β/IL-17 treated osteochondral grafts". In their manuscript the authors focused on the influence of glucocorticoids (GC), hyaluronic acid (HA), and their combination on osteochondral grafts treated with proinflammatory cytokines, assessing their effects on biological and tribological outcome measures.

Overall the topic is very interesting. The manuscript is well written and is scientifically sound. The discussion is based on relatively scarce literature. However, some points must be cleared before the manuscript could be considered for publication.

Specific:
(1) Results.
“Conversely, the GC and GC/HA treated groups revealed a decline in gene expression for the non-cartilage-specific gene COL1A1 (Error! Reference source not found.C) and for the genes encoding the degradative enzymes MMP1 and MMP13 (Error! Reference source not found.D and E).” – Please find and insert the missed references through the manuscript.
(2) Discussion.
Please mention and discus other less invasive options in treating early osteoarthritis (e.g. doi:10.3390/bioengineering10020208).  
(3) Conclusions.
This is the worst part of the manuscript and should be expanded. The authors are encouraged to point out the results of their important study. State clearly, what is making your study unique and special.  

Reviewer 4 Report

This study entitled “The Combination of Glucocorticoid and Hyaluronic Acid Enhances Efficacy on IL-1β/IL-17-Treated Bovine Osteochondral Grafts Compared with Individual Application” seems to have been generally well executed and written. Furthermore, I believe that this paper will be of great interest to the readers. However, I have a few remarks that require authors attention.

Title

Please add the type of article in your title.

Keywords

Consider additional keywords to help readers more easily identify your paper.

4. Materials and Methods

First subsection should be the Study design.
Please begin this subsection with an information what type of study you have performed, in which time period and where. Furthermore, I did not notice an Ethical approval for conducting the study. Therefore, in this subsection of Materials and Methods, please include the information who approved conducting this study, number of Ethical approval, and the date of Ethical approval. Finally, did you register your study on ClinicalTrials.gov or something similar. If yes, please state this information here, with the date and number of the registration.

4.9. Statistical Analysis

Why was the sample size calculation not performed?

3. Discussion
Please begin Discussion with the main findings of your study.

Are there any long-term repercussions on patient health if Hyaluronic Acid injections are given for prolonged time and frequently? Please discuss this issue.

Background and discussion are not sufficiently expanded regarding current evidence in the literature and the number of limitations of the study (e.g., Caric et al, Int J Mol Sci, 2021).

Round 2

Reviewer 2 Report

No further corrections are needed.

regards

Reviewer 3 Report

The authors improved their manuscript substantially, but not completely. 

In the Introduction section the authors stated: "Extensive research has focused on chondroprotective approaches to delay surgery and preserve joint function. While lifestyle changes, physiotherapy, and pain medication are commonly used in mild knee OA, these interventions only provide symptomatic relief without protecting the cartilage[6]. In advanced stages of OA, where cartilage wear is severe, and bone exposure occurs, knee arthroplasty remains the preferred treatment method." From the point of complete cover of the topic of OA treatment, I would strongly suggest to mention the subchondroplasty procedure, which is gaining popularity and hustles between intraarticular injections and arthroplasty. However, the subchondroplasty does not have a cartilage-protection effect as has been suggested in some large animal studies. A critical review on this topic has been published recently ond should be included in the manuscript (doi:10.3390/bioengineering10020208).    

Round 3

Reviewer 3 Report

Well done. No further qerries.